# Reconstruction of Ewing Sarcoma Developmental Context from Mass-Scale Transcriptomics Reveals Characteristics of EWSR1-FLI1 Permissibility

**DOI:** 10.3390/cancers12040948

**Published:** 2020-04-11

**Authors:** Henry E. Miller, Aparna Gorthi, Nicklas Bassani, Liesl A. Lawrence, Brian S. Iskra, Alexander J. R. Bishop

**Affiliations:** 1Department of Cell Systems and Anatomy, University of Texas Health at San Antonio, San Antonio, TX 78229, USA; millerh1@livemail.uthscsa.edu (H.E.M.);; 2Greehey Children’s Cancer Research Institute, University of Texas Health at San Antonio, San Antonio, TX 78229, USA

**Keywords:** Ewing sarcoma, EWSR1-FLI1, transcriptomics, manifold learning, single cell biology, R-loops, replication stress, sarcomagenesis, developmental trajectories, cell identity

## Abstract

Ewing sarcoma is an aggressive pediatric cancer of enigmatic cellular origins typically resulting from a single translocation event t (11; 22) (q24; q12). The resulting fusion gene, *EWSR1-FLI1*, is toxic or unstable in most primary tissues. Consequently, attempts to model Ewing sarcomagenesis have proven unsuccessful thus far, highlighting the need to identify the cellular features which permit stable EWSR1-FLI1 expression. By re-analyzing publicly available RNA-Sequencing data with manifold learning techniques, we uncovered a group of Ewing-like tissues belonging to a developmental trajectory between pluripotent, neuroectodermal, and mesodermal cell states. Furthermore, we demonstrated that EWSR1-FLI1 expression levels control the activation of these developmental trajectories within Ewing sarcoma cells. Subsequent analysis and experimental validation demonstrated that the capability to resolve R-loops and mitigate replication stress are probable prerequisites for stable EWSR1-FLI1 expression in primary tissues. Taken together, our results demonstrate how EWSR1-FLI1 hijacks developmental gene programs and advances our understanding of Ewing sarcomagenesis.

## 1. Introduction

Ewing sarcoma is an aggressive cancer of the bone and soft tissues resulting from the fusion of Ewing sarcoma RNA Binding Protein 1 (*EWSR1*) to Friend leukemia integration 1 transcription factor (*FLI1*) (85%) or E-twenty-six-related (ETS-related) gene (*ERG*) (15%) [1]. Ewing sarcoma’s clinical histopathology is heterogenous with some tumors displaying more mesenchymal or more neuroectodermal characteristics [2]. Given the relative paucity of mutations in these tumors [3], Ewing sarcoma’s heterogeneity is likely determined by the cell from which it arises. However, ectopic EWSR1-FLI1 expression is toxic or unstable in most primary cell types and, to date, no successful model of Ewing sarcomagenesis has been established [4,5,6,7]. These facts highlight the urgent need to understand the contextual requirements for stable EWSR1-FLI1 expression within normal tissue types.

Owing to the recent efforts of Lachmann et al. in reprocessing hundreds of thousands of publicly available transcriptomics datasets, it is now computationally feasible to mine gene expression data at mass-scale from a vast range of tissue types, diseases, and drug conditions [8]. Manifold learning provides a set of tools for analyzing high dimensionality data, revealing both global and local relationships [9]. In this study, we utilized manifold learning approaches to analyze 40,903 bulk transcriptomes of normal tissues and Ewing sarcoma. From this analysis, we elucidated the transcriptomic relationship between Ewing sarcoma and the context of normal developmental tissues from which it likely arises, uncovering cellular features which likely provide permissibility for stable EWSR1-FLI1 expression.

## 2. Results

### 2.1. Ewing Sarcoma Developmental Context Reconstructed from Mass-Scale Transcriptomics

To elucidate the transcriptomic context in which Ewing sarcoma arises, we processed 40,903 bulk transcriptomes of normal tissues and Ewing sarcoma samples obtained from the ARCHS^4^ data repository [8]. First, we implemented a Uniform Manifold Approximation and Projection (UMAP) embedding to reveal the global similarities and differences between the samples in our data set [10]. In coordination with Louvain clustering [11], this approach accurately reconstructed known biological groups, such as hepatic, adipose and immune tissues (Figure 1A and Appendix A).

To uncover the normal tissues which display Ewing-like transcriptomic profiles, we calculated a set of Ewing sarcoma marker genes (Figure 1B) and then determined which normal tissues express these genes at high levels (Figure 1C and Appendix A). 

Ewing sarcoma marker genes were calculated by comparing the gene expression of Ewing sarcoma samples to that of the normal tissue samples. The resulting gene set is highly enriched for direct EWSR1-FLI1 targets (Appendix A) and for genes previously associated with Ewing sarcoma (custom “Ewing sarcoma gene set”, see Methods) (Appendix A), indicating the validity of our approach.

To uncover the most Ewing-like normal tissues, we grouped samples by tissue type (Figure 1C) and cluster membership (Appendix A) and ranked them by median Ewing sarcoma marker gene expression level (Figure 1C, Appendix A). This analysis demonstrated the similarity of Ewing sarcoma with pluripotent stem-cells (iPSCs and hESCs), neural crest cells, neural progenitors, and multipotent mesenchymal stromal cells (MSCs) among others (Figure 1C). Given that these tissues are involved in known developmental transitions (e.g., gastrulation), we hypothesized that an analysis of Ewing sarcoma within this context would reveal developmental programs involved in Ewing sarcomagenesis. Furthermore, we suspected that this analysis would also elucidate characteristics of normal developmental tissue types which permit the stable expression of EWSR1-FLI1.

Of note, while cluster membership tended to follow tissue annotation, there were notable exceptions relevant to several Ewing-like clusters (Appendix A). This result indicated that Ewing-like transcriptional states might arise within subtypes of otherwise non-Ewing-like tissue categories. Consequently, we defined “Ewing-like normal tissues” based on which samples belonged to the group of top Ewing-like clusters (Appendix A; clusters 17, 16, 24, 2, 19, 4, and 28) rather than the top Ewing-like tissue categories.

While UMAP is highly effective at grouping samples based on transcriptomic similarities, the Potential of Heat-diffusion for Affinity-based Trajectory Embedding (PHATE) algorithm has demonstrated superiority in revealing subtle transitions between cell states [12]. We applied the PHATE algorithm to reveal the location of Ewing sarcoma in the transitions among Ewing-like normal tissues (Figure 2A). When labeled for germ layer lineage, the PHATE embedding reveals Ewing sarcoma’s location within this developmental context (Figure 2B; Appendix A). Interestingly, even when Ewing sarcoma samples were removed from the analysis, the shape of the PHATE embedding was not noticeably altered (Appendix A). Given that PHATE organized Ewing sarcoma cells along a normal developmental transition from pluripotent/neuroectodermal to mesodermal tissues, the possibility was raised that EWSR1-FLI1 controls lineage-specific gene programs that define cellular identity within these progenitor populations.

### 2.2. *EWSR1-FLI1* Expression Levels Determine Ewing Sarcoma’s PHATE within a Developmental Context

It was previously reported that ectopic EWSR1-FLI1 expression induces neuroectodermal marker expression in MSCs [13] and that EWSR1-FLI1 knockdown in Ewing sarcoma cells promotes mesenchymal marker expression [14]. These findings led us to hypothesize that EWSR1-FLI1 activity controls the position of Ewing sarcoma in the pluripotent/neuroectodermal to mesodermal developmental trajectory along PHATE_1 (Figure 2B; Appendix A). Studies which involved depleting EWSR1-FLI1 were analyzed for corresponding changes in PHATE_1 position (Appendix A). It was found in each case that EWSR1-FLI1 knock-down drove cells higher on PHATE_1 towards the mesodermal branch and away from the pluripotency/neuroectodermal lineage branches. This phenomenon is exemplified by a comparison of samples with and without EWSR1-FLI1 shRNA knockdown (transcriptomic data obtained from Howarth et al. [15]; GSE60949) (Figure 3B). To confirm this finding, we first calculated the Pearson correlation of gene expression and PHATE_1 position across Ewing samples, yielding a “PHATE_1 correlation score” (signed R^2^) for every gene. This revealed the genes which drive samples higher on PHATE_1 and vice versa (Figure 3C). After ranking genes by their PHATE_1 correlation score, we were able to determine what pathways were correlated with higher and lower PHATE_1 positions using gene set enrichment analysis (GSEA) [16] (Figure 3D). From this analysis we found that markers of low EWSR1-FLI1 expression were strongly correlated with increasing PHATE_1 scores and vice versa. In agreement with the previous analysis, this result also indicates that the transition from low to high EWSR1-FLI1 expression correlates with the transition from mesodermal to pluripotent/neuroectodermal cell states in normal tissues. This result was further confirmed by GSEA of other pathways correlated with Ewing sarcoma’s position in PHATE_1, using gene sets from the Molecular Signatures Database (MSigDB) Chemical and Genetic Perturbations (C2:CGP) collection [17]. As expected, the correlation of gene expression with PHATE_1 in Ewing cells was significantly enriched for mesenchymal-like cancer pathways (in the case of positive correlations), such as “Verhaak Glioblastoma Mesenchymal”, and pluripotent-like pathways (in the case of negative correlations), such as “Wong Embryonic Stem Cell Core” (Appendix A). These results further confirmed our observation that EWSR1-FLI1 expression pushes cells along an innate developmental trajectory between mesodermal and pluripotent/neuroectodermal cell states. In addition to EWSR1-FLI1 knock-down, there were several other interventions which significantly pushed Ewing sarcoma along this developmental trajectory (Appendix A).

It was previously reported that lysine-specific histone demethylase 1 (LSD1) inhibition disrupts the Ewing sarcoma transcriptome [18]. In agreement with this finding, we found that LSD1-inhibiting interventions like SP2509 treatment and LSD1 knock-down pushed Ewing sarcoma higher on PHATE_1 (Appendix A). The response to LSD1 inhibition was observed in vitro, but, as LSD1 inhibitors are currently being tested clinically for Ewing sarcoma, it remains to be evaluated whether the same response would occur in vivo. Furthermore, recent literature indicates that EWSR1-FLI1 antagonizes TEA domain transcription factor 1 (TEAD1) transcriptional programs [19]. We found that inhibition of TEAD1 pushes Ewing sarcoma lower on PHATE_1, indicating that this antagonism is likely bi-directional (Appendix A).

To test whether Ewing sarcoma’s PHATE_1 gene correlations were distinct from those of the underlying developmental context, these analyses were repeated in the absence of any Ewing samples and the results were compared (Appendix A). Quite surprisingly, a significant overlap in C2:CGP and Ewing sarcoma gene set enrichment was observed between the gene correlations along PHATE_1 calculated from Ewing sarcoma samples and those calculated from the Ewing-like normal tissues (Appendix A). The conservation of Ewing sarcoma pathway enrichment in the transition between normal tissue states provides further confirmation that EWSR1-FLI1 controls the movement of cells along this innate developmental trajectory. Furthermore, the enrichment of Ewing sarcoma gene sets in the transitions among primary tissue types indicates that Ewing sarcoma gene sets are largely markers of cellular identity rather than bona fide markers of Ewing sarcoma.

### 2.3. PHATE_1 Gene Scores Identify Mesenchymal-Like Cellular Subpopulation in Ewing Sarcoma Single Cell Transcriptomes

Recent reports indicate that EWSR1-FLI1 expression levels play a role in defining tumor heterogeneity, particularly in defining proliferative and migratory subpopulations [14,20]. In the above results, we found that EWSR1-FLI1 pushes Ewing sarcoma cells along a developmental trajectory between pluripotent/neuroectodermal and mesodermal cell states. Consequently, we hypothesized that developmental gene expression profiles would also be evident at the single cell level and correlate with markers of EWSR1-FLI1 expression. We generated single cell transcriptomes of Ewing sarcoma cell lines and merged them with recently published single cell profiles of patient-derived xenografts (PDXs) (Figure 4A and Appendix A). Cell cycle states and Louvain clusters were assigned to each cell (Figure 4B,C). Importantly, the proportion of cells in G1 compared to G2 or S phase depended on whether cells were derived from a PDX or cell line (Appendix A).

To uncover cellular subpopulations shaped by developmental gene expression programs, we took the PHATE_1 correlation scores for each gene and used them to weight the gene expression profile of each single cell (see Methods). By calculating the median of these weighted gene expression levels, we assigned a “PHATE_1 score” to each cell. Similar to the bulk analysis, a higher PHATE_1 score indicates a cell has a more mesodermal transcriptome and a lower PHATE_1 score indicates a more pluripotent/neuroectodermal transcriptome. To uncover cellular subpopulations which display evidence of higher or lower PHATE_1 score, we ranked each cluster by the median PHATE_1 scores of the cells which belong to it (Figure 4D).

The top PHATE_1-high cluster (Cluster_8) was further characterized by the identification of positive and negative marker genes via differential gene expression analysis of this cluster compared to all others (Figure 4E; Appendix A). Pathway enrichment was conducted using the positive and negative selection markers and gene sets from multiple MSigDB collections (custom “integrated” gene set collection, see Methods) (Figure 4E; Appendix A). The enrichment of high PHATE_1 scores (mesodermal branch) in this cluster was mirrored by the pathway enrichment for mesenchymal-like gene sets such as “Hallmark Epithelial Mesenchymal Transition”, loss of cell cycle activation indicated by “Iglesias E2f Targets Up (Down)”, and for the EWSR1-FLI1-low gene set “Kinsey Targets of Ewsr1 Fli1 Fusion Down”. These results recapitulated the bulk analysis, finding that EWSR1-FLI1 expression levels control the activation of developmental gene programs and lineage commitment. Furthermore, this analysis demonstrated the relevance of these developmental programs in defining Ewing sarcoma tumor heterogeneity.

### 2.4. Permissible PHATE_1-Low Tissues Show Important Transcriptional Similarities with *EWSR1-FLI1* Transcriptome

EWSR1-FLI1 expression is toxic to most cells into which it is introduced. However, several primary cell types are capable of stable ectopic EWSR1-FLI1 expression: iPSCs, hESCs (only tested with p53 knock-down), neural crest cells, and neural crest-derived MSCs (NC-MSCs) [6,21,22,23]. Interestingly, each of these tissue types (except for NC-MSCs which were not available for analysis) belongs to the PHATE_1-low developmental context (Figure 2A) and display high expression of Ewing sarcoma marker genes (Figure 1C). These findings led us to hypothesize that PHATE_1-low tissues can stably express EWSR1-FLI1 because of their basal similarity with the EWSR1-FLI1 transcriptome. To characterize the shared biological features that could account for permissibility, we performed a comparative pathway enrichment for genes of the EWSR1-FLI1 transcriptome and markers of PHATE_1-low tissues (Figure 5). The “EWSR1-FLI1 Transcriptome” was defined as the genes which are activated by EWSR1-FLI1 or increase after ectopic EWSR1-FLI1 expression (see Methods for additional details). The set of “PHATE_1-low Markers” were defined as the genes with a PHATE_1 correlation score <−0.2 in normal tissues. For both groups, pathway enrichment (gene set overrepresentation) was calculated using a collection of gene sets from MSigDB (“integrated” collection, see Methods For additional details). A highly significant overlap was found (Figure 5A).

The 289 shared gene sets were assigned to general biological categories (see Methods for additional details) and the number of gene sets in each category was compared (Figure 5B). Interestingly, the top results belonged to biological processes related primarily with cell cycle, DNA damage response, and RNA processing.

### 2.5. The Capability to Resolve R-Loops and Replication Stress are Probable Requirements for Stable *EWSR1-FLI1* Expression

Ewing sarcoma cells display high rates of replication and transcription [24], which is unsurprising given their cancer status. However, recent reports indicate that pluripotent tissues also display these characteristics [25,26,27]. Furthermore, high rates of transcription combined with high rates of proliferation can create transcriptional stress and genome instability via the production of R-loops (genomic structures formed by the hybridization of RNA and DNA) [28]. Consequently, we hypothesized that factors which resolve R-loops and replication stress are required to support the high replication and transcription rates resulting from EWSR1-FLI1 expression. If correct, we predict the following should also be true: (1) PHATE_1-low, but not PHATE_1-high, tissues display high levels of R-loop accumulation, (2) that Ewing sarcoma cells depend upon factors which resolve R-loops and replication-stress, and (3) that this dependency is shared by PHATE_1-low tissue types but not PHATE_1-high tissue types.

If our first assertion is correct, then it follows that PHATE_1-low (pluripotent/neuroectodermal) tissues would show significantly higher accumulation of R-loops compared to PHATE_1-high (mesodermal) tissues. We mined publicly available DNA-RNA immunoprecipitation sequencing (DRIP-Seq) data from a recently published gene expression omnibus (GEO) accession (GSE145964; Appendix A) to test this assertion. The data was re-processed and the number of peaks (R-loop sites) in each cell type was compared (Figure 6A). This analysis revealed that there are significantly more R-loops in PHATE_1-low cell types (iPSCs, NSCs, and hESCs) compared to a PHATE_1-high cell type (MSCs). Furthermore, we noted that a large number of DRIP-Seq peaks are found exclusively in iPSCs compared MSCs (Figure 6B).

Second, we predicted that Ewing sarcoma would depend upon R-loop and replication-stress-mitigating factors to maintain its high levels of transcription and proliferation. Previously, we and others noted that Ewing sarcoma cells activate the Ataxia telangiectasia and Rad3 related (ATR) replication stress signaling pathway and that this depended upon the presence of R-loops [24,29]. However, we had not identified any downstream effectors that might be key to processing R-loops and maintaining genome stability in these tumors. By examining the gene correlations along PHATE_1 in Ewing sarcoma cells (Appendix A), we discovered several genes strongly correlated with an EWSR1-FLI1-high cell state which are relevant to R-loop resolution and replication stress: 1) the Fanconi Anemia pathway (FANCI, FANCD2, and FANCA), and 2) Flap endonuclease 1 (FEN1).

Fanconi Anemia genes play a critical role in resolving both interstrand crosslinks and R-loops, supporting replication fork integrity and preserving genome stability [30]. Compared to IMR90, a fibroblast and PHATE_1-high cell type, we observed dramatic increase in the post-translationally modified (activated/ mono-ubiquitinated) forms of FANCD2 and FANCI protein in Ewing sarcoma cells (black triangles, Figure 7A) and a marked increase in cell death with knockdown of multiple Fanconi anemia genes (Figure 7B). These results suggest a dependence on this pathway for survival in Ewing sarcoma but not in IMR90 cells.

FEN1 is an essential enzyme that removes 5’ overhangs during DNA repair and replication [31]. FEN1 has also been implicated in processing R-loops at telomeres to limit telomere fragility [31,32]. This led us to hypothesize FEN1 could be important for resolving R-loops in Ewing sarcoma and, consequently, maintaining high proliferation rates. We experimentally validated that FEN1 was upregulated in Ewing sarcoma cell lines compared to IMR90 (Figure 7C). Furthermore, the chemical inhibition of FEN1 resulted in severe toxicity in Ewing sarcoma, but not in IMR90 (Figure 7D). More importantly, unlike in IMR90, FEN1 inhibition led to further accumulation of R-loops in Ewing sarcoma cells (Appendix A).

Finally, it was asserted that PHATE_1-low tissues, but not PHATE_1-high tissues, would demonstrate a dependency upon the factors which resolve R-loops and replication stress (the Fanconi Anemia and flap endonuclease genes). We demonstrated above that PHATE_1-high cell types do not accumulate high R-loop levels (Figure 6 and Appendix A). We also demonstrated that IMR90, a PHATE_1-high tissue type, does not express high levels of the Fanconi Anemia and flap endonuclease genes (Figure 7A,C) or depend upon them for survival (Figure 7B,D). Therefore, it remained only to determine whether PHATE_1-low tissues demonstrate a dependency upon these factors similarly to Ewing sarcoma. We calculated normalized read counts of Fanconi Anemia and flap endonuclease genes for each sample. The results were grouped by tissue type, and tissues were ordered by median expression of all four genes (Figure 8). As expected, we found that PHATE_1-low tissue types (neural crest cells, neural progenitor/stem cells, iPSCs, and hESCs) showed higher expression of Fanconi Anemia and flap endonuclease genes compared to all other tissues except for Ewing sarcoma and HSCs.

Taken together, these results support the assertion that R-loop and replication stress resolving factors are probably necessary for stable EWSR1-FLI1 expression in primary tissue types.

## 3. Discussion

Elucidating the transcriptomic context in which Ewing sarcoma arises involves both the question of which normal cell states are permissive for stable fusion gene expression and of how tumor cells hijack developmental pathways to maintain proliferation, resist treatment, and metastasize. To address these questions, we undertook an unbiased mass-scale re-analysis of publicly available RNA-Sequencing data sets including 40,643 samples from a wide variety of normal tissue types and 260 Ewing sarcoma samples. By analyzing the marker genes which distinguish Ewing sarcoma from other samples, we identified normal tissues with Ewing-like gene expression patterns (Figure 1C). Interestingly, the tissues which showed Ewing-like transcriptomes belong to developmental transitions such as gastrulation (Figure 2B; Appendix A). This led us to hypothesize that EWSR1-FLI1 reactivates developmental gene programs in normal tissue types as part of Ewing sarcomagenesis.

To further elucidate the transcriptomic landscape within the population of Ewing sarcoma and Ewing-like tissues, we implemented the PHATE data diffusion algorithm [12]. While UMAP offers a global and local perspective on the similarities and differences between various samples in a data set, PHATE is capable of accurately reconstructing the transition states between samples based on their gene expression profiles [10,12]. We demonstrated that PHATE can successfully recapitulate the known biological transitions within the bulk transcriptomes of Ewing sarcoma-like tissues (Figure 2 and Appendix A; Appendix A). This approach was particularly effective in revealing the transition states between pluripotent, neuroectodermal, and mesodermal lineages, revealing the location of Ewing sarcoma within this developmental context (Figure 2B; Appendix A). Though Ewing sarcoma is believed to arise from mesenchymal origins, it also displays markers of pluripotency and neuroectodermal lineages [2]. This dichotomy was recapitulated by PHATE as it portrayed Ewing sarcoma as stretched between the mesodermal branch and the juncture between the pluripotent and neuroectodermal branches (Figure 2B; Appendix A).

Multiple studies have demonstrated that ectopic EWSR1-FLI1 expression leads to the expression of neuroectodermal markers in MSCs and that EWSR1-FLI1 depletion in Ewing sarcoma cells leads to increased mesenchymal gene expression patterns [13,14]. Consistent with these findings, our analysis showed that samples with lower EWSR1-FLI1 expression had higher PHATE_1 positions (more mesodermal), and samples with higher EWSR1-FLI1 expression had lower PHATE_1 positions (more pluripotent/neuroectodermal) (Figure 3 and Appendix A).

Several transcriptomic studies have defined Ewing sarcoma-specific gene sets, many of which have been formalized in the Molecular Signatures Database (MSigDB) Chemical and Genetic Perturbations (C2 : CGP) collection [17]. These gene sets are typically generated through transcriptomic experiments in which a non-Ewing cell is engineered to express EWSR1-FLI1 or a Ewing cell is subjected to knock-down of EWSR1-FLI1 and the change in gene expression is determined. As expected, these gene sets are highly enriched in the gene expression correlations of Ewing sarcoma samples along PHATE_1 (Figure 3D). However, the same is true for the underlying developmental context. Even when no Ewing sarcoma samples are considered, the gene correlations within the developmental context along PHATE_1 are also highly enriched for “Ewing sarcoma gene sets” (Appendix A). These results indicate the degree to which EWSR1-FLI1 hijacks normal developmental trajectories and also indicates that many of the classical Ewing sarcoma gene sets may be, in fact, markers of cell lineage rather than bona fide markers of Ewing sarcoma.

Previous reports demonstrated that inhibition of LSD1 interferes with the EWSR1-FLI1 transcriptomic program [18]. Recapitulating this observation, we demonstrated SP2509 treatment acts similarly to EWSR1-FLI1 knockdown by shifting multiple Ewing sarcoma cell lines significantly higher on PHATE_1 (Appendix A). Furthermore, previous studies have described an antagonistic relationship between TEAD1 and EWSR1-FLI1 in which EWSR1-FLI1 prevents the transcription of TEAD1 target genes, preventing ECM sensing [19]. We demonstrate that TEAD1 siRNA shifts Ewing sarcoma cells significantly lower on PHATE_1, toward the pluripotent/neuroectodermal branch (Appendix A). This finding implies that the antagonism between EWSR1-FLI1 and TEAD1 is bi-directional and that Ewing sarcoma cells can be induced to undergo further transformation in basal conditions through TEAD1 knock down.

Recent studies have demonstrated that EWSR1-FLI1 activity plays a central role in defining Ewing tumor heterogeneity [14,20]. Combining single cell transcriptomes from cell line and PDX samples, we identified a subpopulation which displayed the highest PHATE_1 score (mesodermal). As expected, our cluster marker analysis confirmed that this subpopulation is enriched for mesodermal lineage markers and markers of low EWSR1-FLI1 expression (Figure 4E). Interestingly, this analysis also revealed the degree to which this subpopulation is defined by markers of invasiveness and metastasis. The top positive markers in these cells are S100 Calcium Binding Protein A10 (*S100A10*), secreted Protein Acidic And Cysteine Rich (*SPARC*), and lamin A/C (*LMNA*), all three of which have been firmly linked with metastatic progression in multiple cancers [33,34,35]. Furthermore, pathway enrichment of the markers in this cluster reveals evidence of epithelial to mesenchymal transition and low-EWSR1-FLI1 expression alongside increased expression of multiple metastasis-related pathways (e.g., “WU_CELL_MIGRATION”, *p* < 5.94 × 10^−9^), implicating these cells as a likely metastatic subpopulation (Figure 4E; Appendix A). Recent literature has indicated that EWSR1-FLI1 levels fluctuate within tumors and that this heterogeneity controls the ability of cells to proliferate or metastasize [20]. Our findings reveal that cell proliferation and metastasis are not unique to Ewing sarcoma but are capabilities derived from its hijacking of the transcriptional programs found within its developmental context.

Though many consider bone-marrow MSCs to be the probable cell of origin for Ewing sarcoma, neither they, nor fibroblasts (another PHATE_1-high tissue), are capable of stable EWSR1-FLI1 expression [4,6,23,36]. Conversely, PHATE_1-low tissues iPSCs, hESCs (only tested with p53 knock-down), and neural crest cells each tolerate stable EWSR1-FLI1 expression and, in the case of hESCs, show dependency upon the fusion gene [6,21,22,23]. Given the transcriptional similarity between PHATE_1-low tissue types and EWSR1-FLI1-high Ewing sarcoma samples, we hypothesized that these primary tissues might possess qualities that make them innately permissive for EWSR1-FLI1 expression. Subsequent analysis revealed that PHATE_1-low tissues share a high degree of similarity with the EWSR1-FLI1 transcriptome, largely pertaining to proliferation, DNA repair, and RNA processing pathways (Figure 5).

Previous reports demonstrated that both Ewing sarcoma and pluripotent tissues display high levels of proliferation and transcription [24,25,26]. A known consequence of transcriptional activity is the accumulation of R-loops, three-stranded genomic structures resulting from the hybridization of RNA to DNA [37]. The failure to resolve R-loops leads to replication stress and genomic instability, a condition exacerbated by high replication rates [37]. Ewing sarcoma displays a high abundance of R-loops [24], but it maintains a relatively stable genome [3]. Therefore, we hypothesized that Ewing sarcoma relies upon factors which resolve R-loops and mitigate replication stress. Given the permissivity for EWSR1-FLI1 in PHATE_1-low tissues, we further hypothesized that they would display high R-loop levels and share Ewing sarcoma’s predicted dependency upon R-loop and replication-stress-mitigating factors.

It was experimentally revealed that the Fanconi Anemia and flap endonuclease genes, which have roles in resolving R-loops and mitigating replication stress, are necessary for cell viability in Ewing sarcoma (Figure 7). Furthermore, reanalysis of public DRIP sequencing data revealed that PHATE_1-low tissue types display significantly higher R-loop accumulation compared to PHATE_1-high tissues (Figure 6) and higher expression of Fanconi Anemia and flap endonuclease genes compared to all other tissue types except for Ewing sarcoma and HSCs (Figure 8). Importantly, these experiments also demonstrated that PHATE_1-high tissues which do not permit stable EWSR1-FLI1 expression have (1) lower levels of R-loops compared to PHATE_1-low tissues and Ewing sarcoma (Figure 6 and Appendix A) and (2) low dependence on Fanconi Anemia and flap endonuclease genes (Figure 7 and Figure 8). Taken together, these results indicate that the factors which resolve R-loop accumulation and replication stress, such as the Fanconi Anemia and flap endonuclease proteins, are likely prerequisites for stable EWSR1-FLI1 expression in primary tissues.

There exists a natural antagonism between replication and transcription [27,37]. Replication forks can stall, reverse, or collapse due to collisions with transcriptional machinery [38]. R-loops form naturally during transcription but, if unresolved, contribute to transcription-replication conflicts [39]. Without the ability to mitigate this antagonism, cells cannot maintain hyperproliferation and hypertranscription without suffering genome instability [27]. We demonstrated herein that Ewing sarcoma and PHATE_1-low cell types display the ability to resolve R-loops and replication stress. Consequently, we propose that this mitigating capability is required to maintain the high rates of proliferation and transcription which these tissues generally display (Ewing sarcoma and pluripotent stem cells show both hypertranscription and hyperproliferation) [24,25,26,27]. However, we cannot exclude the possibility that R-loop levels, R-loop resolving factors and factors involved in responding to replication stress are independent correlates of altered proliferative state. It remains now for subsequent experiments to determine whether these mitigating factors can (1) improve the permissibility of other normal tissues for modeling Ewing sarcomagenesis and (2) guide the search for which cell types and cell states Ewing sarcoma could arise from. Furthermore, according to the depmap database, *FEN1* is among the top dependencies for Ewing sarcoma [40]. Taken together with our findings, this supports the concept that Ewing sarcoma cells rely upon factors that mitigate replication-transcription conflict for survival. Consequently, we propose that inhibitors which target these factors may represent robust novel therapeutics for the treatment of Ewing sarcoma patients.

## 4. Materials and Methods

### 4.1. Ewing Sarcoma Cell Lines

Ewing cells TC32, CHLA10, CHLA9, and TC71 were purchased from Children’s Oncology Group (COG) and the EWS502 Ewing sarcoma cell line was a kind gift from Dr. Stephen Lessnick (Nationwide Children’s Hospital, Columbus, OH, USA). IMR90, a primary fibroblast cell line was used as control and purchased from ATCC. TC32, TC71, and EWS502 were grown in RPMI supplemented with 10% Fetal Bovine Serum (Atlanta Biologicals, Flowery Branch, GA, USA), CHLA10 and CHLA9 in IMDM supplemented with 20% fetal bovine serum and IMR90 in DMEM supplemented with 10% fetal bovine serum. Cells were maintained in 37 °C in a humidified atmosphere with 5% CO_2_.

### 4.2. Single Cell RNA-Sequencing (scRNA-Seq) of Ewing Sarcoma Cell Lines

To obtained a single cell suspension, 70–80% confluent cells were washed twice with HBSS (Corning, Corning, NY, USA), recollected using 0.25% Trypsin, 2.21 mM EDTA, 1X sodium bicarbonate solution (Corning) and centrifuged for 5 minutes at 300 rcf at 4 °C. Cell pellets were then resuspended in cold DPBS, without calcium and without magnesium (Corning) supplemented with 0.04% bovine serum albumin (BSA) to minimize cell sticking. Viability was measured using an automated cell counter (Nexcelom, Lawrence, MA, USA), centrifuged as described above and resuspended at the desired final concentration in cold DPBS, without calcium and without magnesium - 0.04% BSA. Finally, single cell suspensions were strain through a 70 μm cell sieve and processed for sequencing.

Library prep was performed with the 10x Genomics v3.1 3’ kit (Pleasanton, CA, USA). All samples were sequenced on the Illumina HiSeq 3000 (San Diego, CA, USA) with a 28 + 8 + 100 configuration. CHLA9 and CHLA10 libraries were re-sequenced on the Illumina NovaSeq (San Diego, CA, USA) with a 28 + 8 + 91 configuration.

### 4.3. Cell Viability Assay

Cells were seeded at 30% confluence in 384 well plates and treated with FEN1 inhibitor (PTPD, Glixx Laboratories Inc., Hopkinton, MA, USA) the following day. Cytotoxicity was evaluated after 72 h of treatment using Celltiter-Glo (Promega, Madison, WI, USA). For transfection experiments, siRNA against FANCI, FANCD2, FANCA and non-targeting Control were purchased from SantaCruz Biotechnology Inc. (Dallas, TX, USA). Cells were incubated with siRNA and Lipofectamine RNAiMax by reverse transfection in 96 well plates, following manufacturer’s instructions. Cell viability was evaluated after 72 h of transfection using Celltiter-Glo.

### 4.4. Western Blot Analysis

Whole cell lysates were prepared using RIPA buffer, separated on 3–8% gradient gels using the NuPage system (Invitrogen, Thermo Fisher Scientific, Waltham, MA, USA) and transferred onto nitrocellulose membrane. Blots were incubated with 1:1000 dilution of the antibodies overnight at 4 °C and developed using enhanced chemiluminescence (Super ECL or West Femto, Thermo Fisher Scientific). The following antibodies were used: FEN1 (sc-13051, Santa Cruz), FANCA (sc-28215, Santa Cruz, Dallas, TX, USA), FANCD2 (NB100-182, Novus, Wrentham, MA, USA), FANCI (A301-254A, Bethyl Labs, Montgomery, TX, USA), β-Tubulin (cs2128, Cell Signaling, Danvers, MA, USA), Vinculin (cs13901, Cell Signaling), goat anti-mouse IgG-HRP and goat anti-rabbit IgG-HRP (Santa Cruz Biotech Inc.). Full images of blots are available in the Appendix A.

### 4.5. RNA:DNA Hybrid Intensity Analysis

Extent of genomic R-loops in the cells was measured using dot blots. Briefly, DNA harvested and purified by phenol-chloroform-ethanol method was digested using a cocktail of restriction enzymes (HindIII, EcoRI, BsrGI, XbaI and SspI, NEB). Each sample (0.5 µg of purified digested genomic DNA) was loaded in quadruplicate on to a pre-wet H^+^ nylon membrane (Amersham Hybond, GE Healthcare Life Sciences, Chicago, IL, USA) and allowed to incubate for 20 min. Membranes were then washed twice with dH_2_O, rinsed in 2× SSC buffer and then left to air-dry at room temperature. For single-stranded DNA, there was an additional denaturation step (incubation in 0.5N NaOH, 1.5M HCl for 10 min), followed by a 10 min incubation in neutralization buffer (1M NaCl, 0.5M Tris-HCl pH 7). Membranes were blocked with 5% non-fat dry milk in TBS-T and incubated with either S9.6 antibody (ENH001, 1:1000, Kerafast, Boston, MA, USA) or ssDNA antibody (MAB3034 1:5000 Millipore, Burlington, MA, USA) overnight. Blots were analyzed with enhanced chemiluminescence (Super ECL) using Image Studio (LI-COR, Lincoln, NE, USA) to measure signal intensity of each dot. The corresponding ssDNA signal was used to normalize R-loop intensity. Cells were treated with a FEN1 inhibitor (RF00974SC, Maybridge Chemicals, Loughborough, UK).

### 4.6. Gene Set Collections Generated and Utilized

Gene sets used within this study for gene set enrichment analysis (GSEA) and gene set enrichment (over-representation analysis) were obtained from the Molecular Signatures Database (MSigDB) v7 accessed via the *msigdbr* function of the *msigdbr* package v.7.0.1 or generated from literature sources. A custom gene set category, termed “integrated”, was taken as the combination of gene sets from the “H” (Hallmark), “C2 : CGP” (chemical and genetic perturbations), “C2 : CP” (Canonical pathways; includes *BioCarta*, *KEGG*, *PID*, and *Reactome*), and “C5 : GO” (Gene Ontology) collections. Additionally, the custom collection of “Ewing sarcoma gene sets” was curated in a three-step process: (1) Five new gene sets were defined from recent literature: “Riggi 2014 activated by EWSR1-FLI1” and “Riggi 2014 repressed by EWSR1-FLI1” were defined from Riggi et al. 2014 as the genes which are bound by EWSR1-FLI1 and transcriptionally activated or repressed based on knockdown experiments in Ewing sarcoma cell lines [41]. “Aynaud 2020 EWSR1-FLI1-high markers”, and “Aynaud 2020 EWSR1-FLI1-low markers” were defined from Aynaud et al. 2020 from independent component analysis of single cell RNA-Sequencing data in a Ewing sarcoma cell line with an inducible knockdown of EWSR1-FLI1, representing gene expression profiles specific to Ewing sarcoma cells with high or low expression of EWSR1-FLI1 respectively [14]. “Aynaud 2020 activated by EWSR1-FLI1” was taken as the intersection of “Aynaud 2020 EWSR1-FLI1-high markers” with the set of genes that show increasing H3K27ac signal in the days following the cessation of full EWSR1-FLI1 knockdown [14]. (2) Pre-existing gene sets related to Ewing sarcoma were gathered from the C2 : CGP collection of MSigDB using the regex term “EWING|_EWS|EWSR1”. (3) Six gene sets of the resulting collection were censored. “Rorie targets of EWSR1-FLI1 fusion down” and “Rorie targets of EWSR1-FLI1 fusion up” involved the expression of EWSR1-FLI1 in neuroblastoma cells and were, therefore, not relevant to a study with a normal tissue background [42]. “Torchia targets of EWSR1-FLI1 fusion up”, “Torchia targets of EWSR1-FLI1 fusion top 20 up”, “Torchia targets of EWSR1-FLI1 fusion down”, and “Torchia targets of EWSR1-FLI1 fusion top 20 down” were conducted in a mouse model of leukemia, which was deemed unsuitable because of the need for a normal tissue background and the divergent genetic background of mice from humans [43]. Furthermore, the set of “EWSR1-FLI1 targets” from Appendix A was taken as the genes from “Ewing sarcoma gene sets” belonging to gene sets which involved an EWSR1-FLI1 or FLI1 ChIP-Seq binding study. Finally, meta gene sets were calculated as part of the analysis in Figure 5 by using regular expressions relevant to the category in question to classify gene sets from the “integrated” collection. The “EWSR1-FLI1 transcriptome” was defined as the “Ewing sarcoma gene sets” which comes from studies that observe changes in gene expression after ectopic expression of EWSR1-FLI1 in non-Ewing tissues or after knock-down of EWSR1-FLI1 in Ewing sarcoma. These gene sets were found through a regex search using the term “_UP|ACTIVATED|HIGH” in the Ewing sarcoma gene sets collection.

### 4.7. Pre-Processing of Publicly Available Bulk RNA-Seq Data

Standardized bulk RNA-Seq data was downloaded from the ARCHS^4^ data repository v8 (February 2020). Sample metadata was categorized by tissue type and tumor status with a custom regex dictionary in coordination with cell line information downloaded from Cellosaurus v33 (December 2019). Samples from single cell experiments were removed. Tumor samples that did not belong to the ‘ewing sarcoma’ category, didn’t fit a unique tissue type classification, or which contained fewer than 10 million reads aligned to the human transcriptome were also filtered out. The result was 40,643 samples from normal tissues and 260 Ewing sarcoma (EWS) samples (Appendix A). Furthermore, genes with zero counts in 10% or more of samples were discarded.

Using the vst function from DESeq2 v1.26.0 counts were geometric-mean normalized and a variance-stabilizing transform was applied, generating a homoscedastic dataset suitable for dimensionality reduction. The resulting normalized gene count matrix had the dimensions 28,621 genes X 40,903 samples. VST-transformed gene variance was calculated for every gene using the rowVars function from the matrixStats v0.55.0 package. The top 10 thousand variable genes were selected for principle component analysis (PCA) calculated with the prcomp function from the stats v3.6.2 package.

The top 100 principle components (PCs) were subsequently used to construct a nearest neighbor graph using the FindNeighbors function of the Seurat v3.1.3 package. Louvain clustering was also performed using the FindClusters function of Seurat and the umap function of the uwot v0.1.5 package was used to calculate a UMAP embedding (Appendix A). Ewing sarcoma marker genes were obtained by using the FindMarkers function of Seurat using the Wilcoxon rank sum model. Only genes with an average log2 fold change >0.58 and p adjusted value <1E-50 were considered as Ewing sarcoma marker genes (Appendix A). Ewing sarcoma marker scores for each sample were calculated as the median of VST-transformed counts for the Ewing sarcoma marker genes within that sample (Appendix A). Additionally, the expression of Fanconi Anemia and flap endonuclease genes across tissue types was taken as the VST-transformed read counts of each gene within each sample (Appendix A). Additionally, comparison of Ewing marker expression between tissue categories and Louvain clusters (Appendix A) was performed using the pairwise.t.test function of the stats R package with ‘holm’ multiple testing correction applied. The enrichment of tissue type within each cluster was calculated as the number of samples of that tissue type in the cluster divided by the total number of samples of that tissue type and then scaled but not centered for each column of the heatmap (Appendix A).

### 4.8. PHATE Analysis of Top EWS-Like Clusters

The top Louvain clusters by Ewing sarcoma marker gene expression were selected for further analysis via the PHATE embedding algorithm [12]. PHATE embeddings were calculated using the phate function of the phateR v1.0.0 package on the matrix of VST-transformed counts. PHATE was calculated for 3 dimensions at a lower nearest-neighbor setting to improve the ability to visualize and remove samples which did not participate in the EWS-containing trajectory (plots generated by accompanying scripts). A final PHATE embedding was optimized from the remaining samples by iterating the nearest neighbor parameter until the developmental context was fully visible (Appendix A).

PHATE-correlated genes were calculated using the cor function of the stats R package (Appendix A). For this analysis, data were “median-by-ratio” normalized using MedianNorm and GetNormalizedMat functions of the EBSeq v1.26.0 package. Gene Set Enrichment Analysis (GSEA) was calculated using the fgsea function of the fgsea v1.12.0 package by using the signed R^2^ of gene correlations along PHATE_1 as the gene ranking metric (Appendix A). The impact of various interventions on PHATE_1 position in Ewing sarcoma samples was calculated using a *t* test implemented via the stat_compare_means function of the ggpubr package. The *t* test was one-tailed in the case of EWS-FLI1 knock-down experiments, and two-tailed in the case of other interventions.

### 4.9. Processing of Ewing Sarcoma Cell Line and Publicly Available scRNA-Seq Data

For Ewing sarcoma cell line scRNA-Seq data, reads were demultiplexed using bcl2fastq (HiSeq 3000-generated reads) and the demux command of the internal 10x genomics pre-processing pipeline (NovaSeq-generated reads). Reads were subsequently standardized to the output style of bcl2fastq. Read counts were produced using the cellranger v3.1 pipeline with the count command and the built-in hg38 reference.

For each dataset, quality control cell filtering was performed based on three metrics calculated by Seurat: (1) Percent mitochondrial reads, (2) number of UMI counts, and (3) number of unique genes identified. Violin plots were used to find thresholds and statistical outliers were automatically removed (plots generated via accompanying scrips). Following quality control, read counts were log normalized and scaled using the NormalizeData and ScaleData functions of Seurat respectively. Then, principle component analysis (PCA) was calculated using the RunPCA function of Seurat. A nearest-neighbor graph and Louvain clustering were calculated using the FindNeighbors and FindClusters functions of Seurat respectively. A UMAP embedding was calculated with the RunUMAP function of Seurat. Cell cycle scoring was performed using the CellCycleScoring function of Seurat. Marker genes were calculated using the FindAllMarker function of Seurat (Appendix A). Pathway enrichment of marker genes for each cluster was calculated using the enricher function of clusterProfiler with annotations taken from the “integrated” gene set collection described above (Appendix A). Word clouds were produced using the wordcloud function of the wordcloud package using integer ranking of marker false-discovery rate within each cluster as the “frequency” metric.

Data set integration was performed using canonical correlation analysis (CCA) in the manner described previously by the authors of Seurat (Appendix A) [44]. First, each sample was separately normalized and the top 5000 highly variable genes were determined using the NormalizeData and FindVariableFeatures functions of Seurat respectively. Then, the FindIntegrationAnchors and IntegrateData functions of Seurat were called to generate an integrated count matrix for downstream analysis via the pipeline described above. PHATE scores were calculated at the single-cell level by multiplying the scaled single-cell gene counts by the corresponding PHATE_1 signed R^2^ gene correlation value obtained from bulk RNA samples. This process weighted each single cell transcriptome by the contribution of the developmental context’s PHATE_1 signature. To obtain a PHATE score for each single cell, the median of all non-zero weighted gene counts was calculated. The resulting cell metadata with PCA, UMAP, and PHATE analysis results is made available here (Appendix A).

### 4.10. Processing of ssDRIP-Seq Data from Developmental Context Cell Types

ssDRIP-Seq (strand-specific DNA : RNA immuno-precipitation sequencing) data was obtained from a recently published public repository (GSE145964, Appendix A). First, raw reads were downloaded in SRA format using the prefetch command from SRA-toolkit v2.10.1. Then the parallel-fastq-dump package/command v0.6.3 was used to split the SRA files into fastq files which were subsequently trimmed and filtered using the fastp package/command v0.20.0. Then the reads were aligned to GRCh38 using the bwa mem command of the bwa package v0.7.17. Alignment files were split by first-in-pair strand, converted to BAM format, sorted, and indexed using samtools v1.9. Strand-specific broad peaks were called using macs2 v2.2.6 with the settings “—nomodel—extsize 147—broad“. BAM reads were assigned to peaks to construct a peak-count matrix for the forward-strand reads and perform PCA using the DiffBind v2.14.0 package. Peaks from batch 2 were removed due to the technical effects discovered by PCA (plots generated via accompanying scripts). Remaining peaks were filtered out if they had a q value >0.001 and were subsequently compared to determine if the number of called peaks was different between cell types using a one-tailed *t* test with an alternative hypothesis of “greater” implemented via the stat_compare_means function of the ggpubr package. Finally, the peaks which passed q value filtering were overlapped to create a consensus peak set for each cell type using the findOverlapsOfPeaks function of the ChIPpeakAnno package v3.20.0. Then the consensus peak sets for iPSCs and MSCs were overlapped and plotted.

### 4.11. Data and Software Availability

All sequencing datasets generated in this study have been deposited in GEO (GSE146221). Publicly available data sets used in this study are listed in Appendix A (RNA-Seq) and Appendix A (DRIP-Seq). All scripts and data used to generate the bioinformatics analyses in this paper have been made publicly available on GitHub (millerh1/Ewing-sarcoma-paper-Miller-2020). Analyses were performed using the R computing language v3.6.2 [45].

## 5. Conclusions

Recent years have yielded powerful insights into the molecular mechanisms by which EWSR1-FLI1 controls gene expression and cellular behavior. However, it is still unclear what cellular characteristics are required for stable EWSR1-FLI1 expression in primary tissues. Consequently, successful spontaneous models of Ewing sarcoma do not yet exist. To address this gap in knowledge, we mined publicly available transcriptomic profiles of Ewing sarcoma and normal tissue types. This allowed us to identify a population of Ewing-like normal tissues belonging to developmental processes such as gastrulation. Additional analysis and experimental validation revealed stable expression of EWSR1-FLI1 likely requires factors which resolve R-loops and replication stress. Taken together, our findings suggest that Ewing sarcomagenesis likely occurs in tissues which already tolerate high proliferation and transcriptional activity because of their capacity for resolving R-loops and mitigating replication stress.

## Figures and Tables

**Figure 1 cancers-12-00948-f001:**
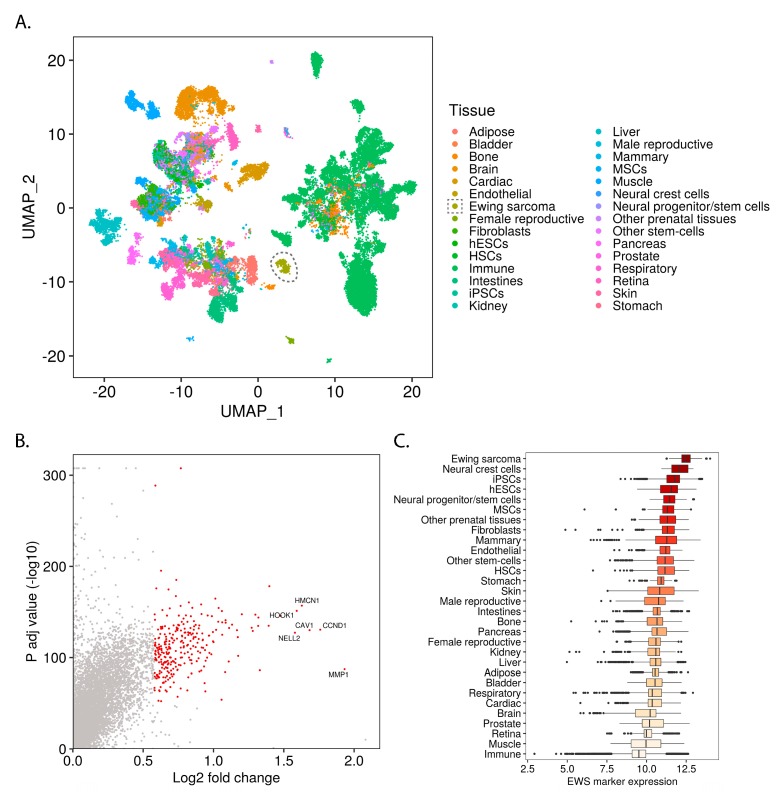
Mass-scale mining of bulk transcriptomes reveals the relationship of Ewing sarcoma to normal tissue types: (**A**) UMAP embedding of bulk transcriptomic profiles from normal tissues and Ewing sarcoma samples (Ewing samples circled); (**B**) One-way volcano plot showing the top Ewing sarcoma marker genes from Wilcoxon rank-sum testing with Bonferroni correction; (**C**) Box-plot comparing the Ewing sarcoma marker gene expression levels for different samples (median of variance stabilizing transform (VST) transformed and geometric mean normalized read counts for all Ewing sarcoma marker genes within each sample), grouped by tissue and ordered by median expression.

**Figure 2 cancers-12-00948-f002:**
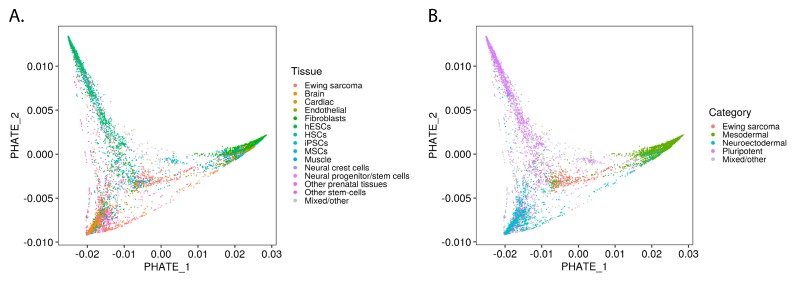
Ewing sarcoma developmental context reconstructed from bulk transcriptomes of Ewing-like tissues: (**A**) PHATE embedding of top Ewing-like samples; (**B**) PHATE embedding labeled by developmental lineage.

**Figure 3 cancers-12-00948-f003:**
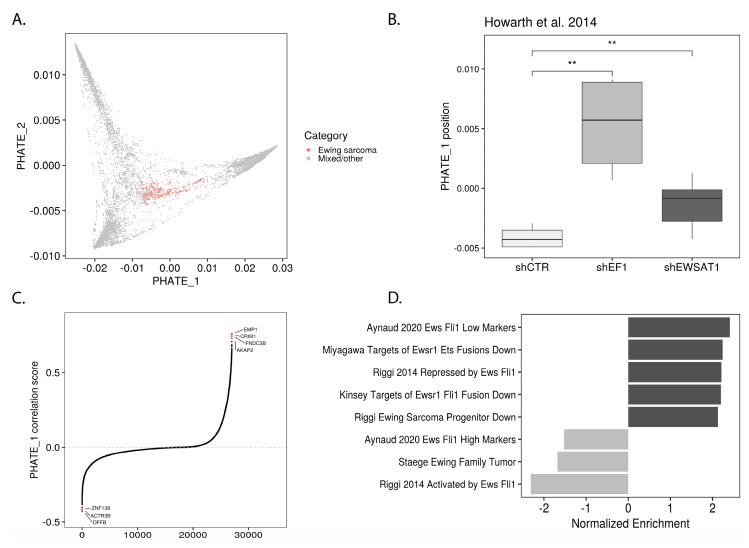
Ewing sarcoma’s position in underlying developmental trajectory controlled by EWSR1-FLI1 expression levels: (**A**) PHATE embedding with Ewing sarcoma samples highlighted; (**B**) Box-plot showing difference in location along PHATE_1 between A673 cells exposed to control shRNA or shRNA targeting EWSR1-FLI1 (shEF1) and Ewing sarcoma associated transcript 1 (EWSAT1) [15] (one-tail *t* test, ** *p* ≤ 0.01); (**C**) Genes in Ewing sarcoma samples ranked by PHATE_1 correlation score (signed R^2^); (**D**) Bar-plot showing enrichment of Ewing sarcoma gene sets within PHATE_1 correlation scores as determined by GSEA.

**Figure 4 cancers-12-00948-f004:**
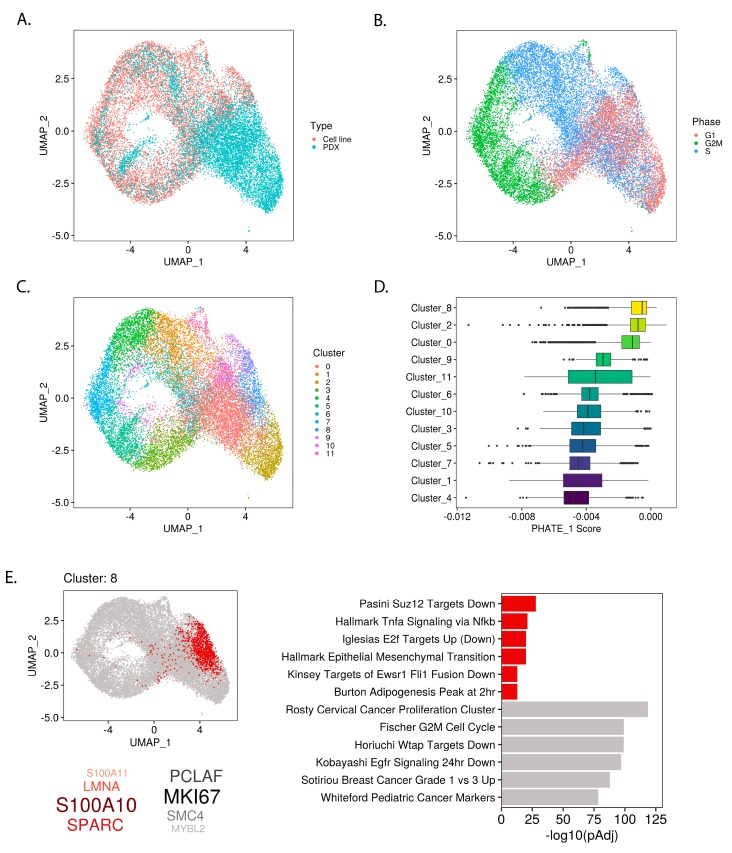
Mesodermal-like subpopulation within single-cell Ewing sarcoma transcriptomes revealed by PHATE_1 expression scores: (**A**) UMAP embedding of single-cell Ewing sarcoma transcriptomes in cell lines and patient-derived xenografts (PDXs) after alignment; (**B**) UMAP showing cell-cycle phase of each cell imputed from gene expression data and (**C**) Louvain clustering assignments; (**D**) Box-plot showing the PHATE_1 expression scores for each sample in each cluster, organized by median score; (**E**) Cluster marker analysis of cluster 8 shows location within UMAP embedding, the top positive marker genes (red) and negative marker genes (grey), and top enriched pathways (with the “integrated” gene set collection) in positive and negative marker genes (red and grey respectively).

**Figure 5 cancers-12-00948-f005:**
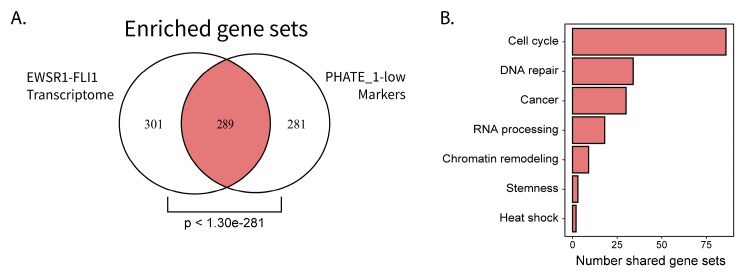
Comparative pathway enrichment reveals potential characteristics of EWSR1-FLI1 permissibility: (**A**) Venn diagram comparing pathway enrichment results for EWSR1-FLI1 transcriptome and PHATE_1-low gene markers (hypergeometric test p value displayed); (**B**) Bar plot showing number of shared gene sets assigned to integrative biological categories.

**Figure 6 cancers-12-00948-f006:**
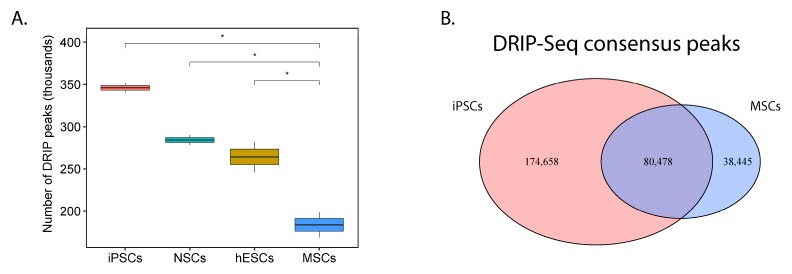
R-loop accumulation is a shared feature of both Ewing sarcoma and its PHATE_1-low developmental context: (**A**) Box-plot comparing number of detected DRIP-Seq peaks (R-loop sites) between cell types found in Ewing sarcoma’s developmental context (iPSCs, induced pluripotent stem cells; NSCs, neural stem cells; hESCs, human embryonic stem cells; MSCs, multipotent mesenchymal stromal cells) (one-tail *t* test, * *p* ≤ 0.05); (**B**) Venn diagram comparing DRIP-Seq peaks in iPSCs compared to MSCs.

**Figure 7 cancers-12-00948-f007:**
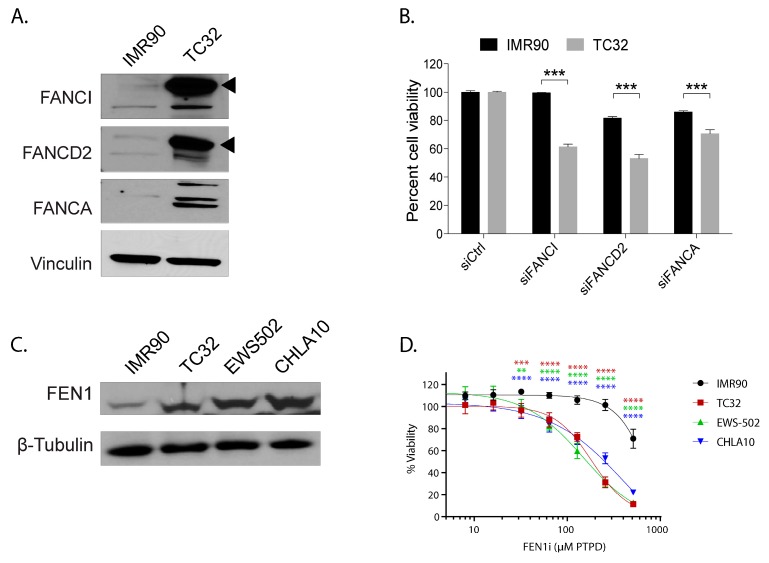
The response to replication stress is a defining feature for Ewing sarcoma compared to the PHATE_1-high developmental context. (**A**) Western blot showing expression of Fanconi Anemia proteins FANCA, FANCD2, and FANCI in TC32 (Ewing sarcoma cell line) compared to IMR90 (fibroblast cell line); (**B**) Bar-plot showing the impact of FANC genes knockdown on cell viability in TC32 compared to IMR90; (**C**) Western blot showing the expression of FEN1 in Ewing sarcoma cell lines TC32, EWS502, and CHLA10 compared to IMR90; (**D**) Line plot showing the viability of Ewing sarcoma cells compared to IMR90 with increasing doses of FEN1 inhibitor. (** *p* ≤ 0.01; *** *p* ≤ 0.001; **** *p* ≤ 0.0001).

**Figure 8 cancers-12-00948-f008:**
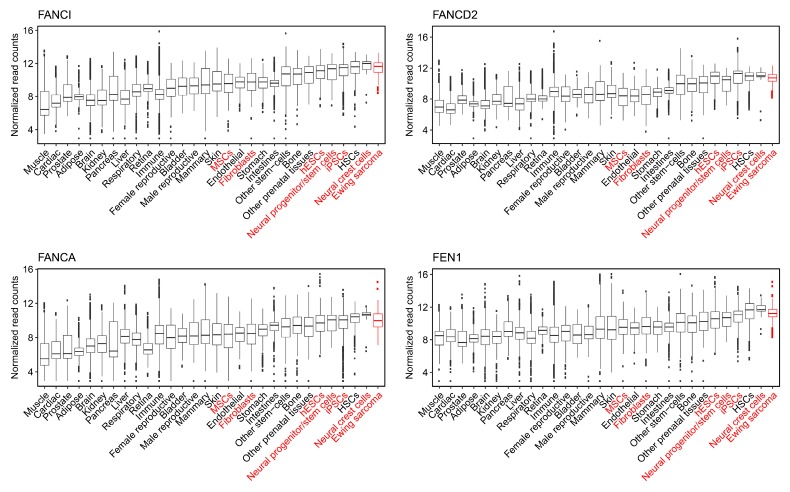
Normalized expression of Fanconi Anemia and flap endonuclease genes across normal tissue types and Ewing sarcoma.

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
