# Peer review of "Reconstruction of Ewing Sarcoma Developmental Context from Mass-Scale Transcriptomics Reveals Characteristics of EWSR1-FLI1 Permissibility"

_cancers, 2020, doi:10.3390/cancers12040948_

Round 1

Reviewer 1 Report

This well-written paper is an exploration of over 40,000 previously published trancriptomes from both normal tissues and Ewing sarcoma, elucidating similarities between the two and arriving at a common, potentially permissive pathway in developmental programming that leads to stable EWSR1-FL1 expression, the recreation or alteration of which may have future therapeutic applications. The authors utilize a variety of techniques to arrive at their conclusions, including Uniform Manifold Approximation and Projection (UMAP) embedding, Potential of Heat-diffusion for Affinity-based Trajectory Embedding (PHATE) embedding, gene set enrichment analysis (GSEA), single-cell transcriptomic profiling, and DRIP-seq analysis. They identify several novel findings that heighten the impact of this publication, including the discovery that EWSR-FL1 expression dictates lineage commitment (mesodermal or pluripotent/neuroectodermal) in both Ewing sarcoma cells as well as the normal pleuripotent stem cells that most closely resemble Ewing Sarcoma. They posit the novel concept that previously identified Ewing sarcoma gene sets may instead be markers for alteration or determination of cellular identity. This also appears to be confirmed by analysis of previously published studies of LSD-1 inhibitor SP2509, and other methods of modulating EWSR1-FLI1's effect. Moreover, the authors establish an empirically derived EWSR-FLI1 transcriptome and are able to determine which genes (Fanconi Anemia and flap endonuclease genes) are important for R-loop resolution and replication stress, the resolution of which may contribute to cell stability and viability. This could form targets for future therapeutics against this disease or create better preclinical models to work with.

Minor Comments
-Line 62 - It is unclear from the description of Figure 1C or from the Methods section how the EWS marker expression number was calculated - it appears that 10.0 units correlates to minimal expression, and 12.5 units to maximal expression, but the meaning of this could be clarified. The differences between the top-most and bottom-most histologies, at first glance, do not appear to be statistically significant.
-Line 122 - It may be helpful to specify that these experiments were done in vitro, as the compound is currently in clinical study
-Line 230 - For Figure 6A, it would be helpful to specify the use of cell lines not mentioned elsewhere (VECs, VSMCs) and why they were used.
-Line 271 - Figure 7's results may be given more explanation here.

Reviewer 2 Report

In the manuscript entitled “Reconstruction of Ewing sarcoma developmental context from mass-scale transcriptomics reveals characteristics of EWSR1-FLI1 permissibility” Miller et al re-analyzing publicly available RNA-Sequencing data with manifold learning techniques, and  uncovered a group of Ewing-like cells belonging to a developmental trajectory between pluripotent, neuroectodermal, and mesodermal cell states. They also propose that permissibility to EWRS1-FLI1 expression depends on cell’s ability to resolve R-loops.

The study addresses an important question in Ewing sarcoma: identifying potential cells of origin of the disease and understanding the context permissible to EWSR1-FLI1 driven oncogenesis. Although the results are not absolutely novel or surprising (the original for Ewing sarcoma has been proposed to be neural crest progenitors rather than MSCs) and it remains to be proven that a particular progenitor can serve as the cell of origin, this study unravels important connections and cues and a similar strategy could be applied to other sarcoma sub-types with yet uncharacterized cell of origin.

Some minor comments:

  1. It is unclear what information is provided by cluster membership in Figure S1B. Do this clusters also correspond to specific tissue types, and which?

  1. It is impossible to read the gene names in Figure 3C.

  1. The single cell analysis suggests the presence of a mesenchymal-like population of cells that is not dividing, and is more similar to the “EWSR1-FLI down” profiles. Are the authors able to separate these cells based on EWSR1-FLI1 expression (if detected in single cell analysis)? In other words, can you show that the heterogeneity observed in cell lines and PDXs is a consequence of fluctuations in the levels of the fusion?

  1. The relationship between the ability to resolve R-loops and permissibility to EWRS1-FLI1 expression is interesting. However, it is still possible that the correlations observed are a consequence of different proliferation rates of the cells in question. For example, R-loops accumulate in cells with high proliferative states (hESC, iPSCs)  which also exhibit high levels of Fanconi anemia factors or other R-loop resolving factors. Cells with lower proliferation rates (IMR90, MSCs) exhibit overall lower levels of these factors, potentially because they are expressed in a cell cycle dependent manner.  In other words, the relationship identified here could be a consequence of proliferation rates rather than a cell intrinsic or identity state, where the ability to resolve R-loops is higher. The authors should discuss this matter.

  1. Line 247: The authors should change “dramatic hyperactivation of FANCD2” to dramatic increase in FANCD2 protein levels, as the figure shows protein levels rather than enzyme activity.

Reviewer 3 Report

This is a well designed scientific study in which the authors have reanalyzed publicly available RNA seq data to better understand the Ewing sarcomagenesis. The methods are sound. The results support the conclusions.

The findings confirm the pleuripotent nature and neuroectodermal origin of Ewing's sarcoma. The figures are well designed but could be of better resolution. The clinical implications and future directions given this finding could be expanded on in the discussion section.

All in all a very well written study.
